# Starting the SToP trial: Lessons from a collaborative recruitment approach

Tracy McRae[1,2]☯*, Roz Walker[1,3,4]☯, John Jacky[3]☯, Judith M. Katzenellenbogen[1]‡, Juli Coffin[2]‡, Ray Christophers[5]‡, Jonathan Carapetis[1,2,6,7], Asha C. Bowen[1,2,6,7]‡

1 University of Western Australia, Perth, Western Australia, Australia, 2 Wesfarmers Centre for Vaccines and Infectious Diseases, Telethon Kids Institute, Nedlands, Western Australia, Australia, 3 School of Indigenous Studies, Poche Centre for Indigenous Health Research, University of Western Australia, Perth, Australia, 4 Ngangk Yira Institute for Change, Murdoch University, Perth, Australia, 5 Nirrumbuk Environmental Health and Services Pty Ltd, Broome, Western Australia, Australia, 6 Perth Children's Hospital, Nedlands, Western Australia, Australia, 7 Menzies School of Health Research, Tiwi, Northern Territory, Australia

☯ These authors contributed equally to this work.
‡ JMK, JC, RC and ACB also contributed equally to this work.
* tracy.mcrae@telethonkids.org.au

## Abstract

### Objective

Recruitment in research can be challenging in Australian Aboriginal contexts. We aimed to evaluate the SToP (See, Treat, Prevent skin infections) trial recruitment approach for Aboriginal families to identify barriers and facilitators and understand the utility of the visual resource used.

### Methods

This qualitative participatory action research used purposive sampling to conduct six semi-structured interviews with staff and five yarning sessions with Aboriginal community members from the nine communities involved in the SToP trial that were audio recorded and transcribed verbatim before thematic analysis.

### Findings

Community members valued the employment of local Aboriginal facilitators who used the flipchart to clearly explain the importance of healthy skin and the rationale for the SToP trial while conducting recruitment. A prolonged process, under-developed administrative systems and stigma of the research topic emerged as barriers.

### Conclusion

Partnering with a local Aboriginal organisation, employing Aboriginal researchers, and utilising flip charts for recruitment was seen by some as successful. Strengthening governance with more planning and support for recordkeeping emerged as future success factors.

**Data Availability Statement:** All relevant data are within the paper and its Supporting Information files.

**Funding:** TM - PhD scholarship from the Australian Centre for Elimination of Neglected Tropical Diseases (ACE-NTD), an NHMRC centre of excellence (APP1153727). ACB - National Health and Medical Research Council [NHMRC] (project grant 1128950) https://www.nhmrc.gov.au/ ACB - Health Outcomes in the Tropical NORTH [HOT NORTH 113932] (Indigenous Capacity Building Grant) https://ddec1-0-en-ctp.trendmicro.com:443/wis/clicktime/v1/query?url=https%3a%2f%2fwww.hotnorth.org.au&umid=0c7bb3ea-de8f-4107-a56a-9dcefed3911a&auth=bd49bbd20ffeb7d6acc8e9a85bb2e9a8f7a67034-16434ef127d1a9508ee61ec5cfbbeb3a52e856c9 ACB - Western Australia Government Healthway Grant 33088 https://www.healthway.wa.gov.au/our-funding/ ACB - NHMRC investigator Award (1175509) https://www.nhmrc.gov.au/ The funders had no role in study design, data collection and analysis, decision to publish, or preparation of the manuscript.

**Competing interests:** The authors have declared that no competing interests exist.

## Implications for public health

Our findings validate the importance of partnership for this critical phase of a research project. Recruitment strategies should be co-designed with Aboriginal research partners. Further, recruitment rates for the SToP trial provide a firm foundation for building partnerships between organisations and ensuring Aboriginal perspectives determine recruitment methods.

## Introduction

Remote living Australian Aboriginal children have the highest reported rate of skin infections globally with almost half the children (45%) suffering from skin sores at any one time [1]. Impetigo (skin sores) is a highly contagious, antibiotic treatable bacterial skin infection following minor trauma, scabies, and insect bites where the skin breaks allow entry of aggressive bacteria [2]. Scabies is a skin infection caused by mites that are transmitted between humans through skin-to-skin contact [3]. Itching and scratching of scabies facilitates the introduction of the bacteria with resultant impetigo [4].

Environmental and social determinants contribute to infectious diseases including skin infections [5, 6], particularly tropical and humid environments, poverty, and limited access to resources. Poor housing hardware [6], limited bedding supplies, overcrowding and normalisation [5, 7] of skin infections all contribute to the burden. To address this, in the Australian Aboriginal context, a comprehensive transdisciplinary approach is required while privileging the knowledge of community members [8]. Acknowledging the enduring effect of colonialism on health and wellbeing through the loss of culture and identity, land dispossession and racism is particularly important [8–11].

This research project is embedded within the See Treat Prevent (SToP) trial, a stepped-wedge cluster randomised trial involving four community clusters in the Kimberley region of Western Australia (WA) aimed at reducing the burden of skin infections by 50% in Aboriginal children living remotely [12]. Embedded within a Community Participatory Action Research (CPAR) approach [13] the trial focuses on co-designing resources, project governance and data sovereignty to ensure Aboriginal people govern the collection, ownership, and application of data about their lands, communities and resources [14]. During the development phase, it was agreed by SToP trial partners Kimberley Aboriginal Medical Service, Western Australia Country Health Service–Kimberley, and Nirrumbuk Environmental Health Services that a visual resource would be used in recruitment to explain about skin infections and the rationale for the SToP trial.

The elements of the SToP trial (Fig 1) include: community consultation and engagement, recruitment, school-based surveillance, and clinic-based treatment of skin infections; and community-driven health promotion and environmental health prevention initiatives.

Indigenous people have arguably been the most researched people in the world with little opportunity to provide input into what, why and how the research is conducted [15]. In an Australian context, this has frequently resulted in feelings of disempowerment where Aboriginal people's needs and priorities have been ignored, with very few improvements in their health outcomes [8–11, 15]. However, there is a strong evidence base that Aboriginal people are willing to participate if they perceive the research privileges Aboriginal voices, priorities, recognises their distinctive experiences and is beneficial for the community as a collective [16]. This requires adopting research methodologies that facilitate meaningful partnerships, two-

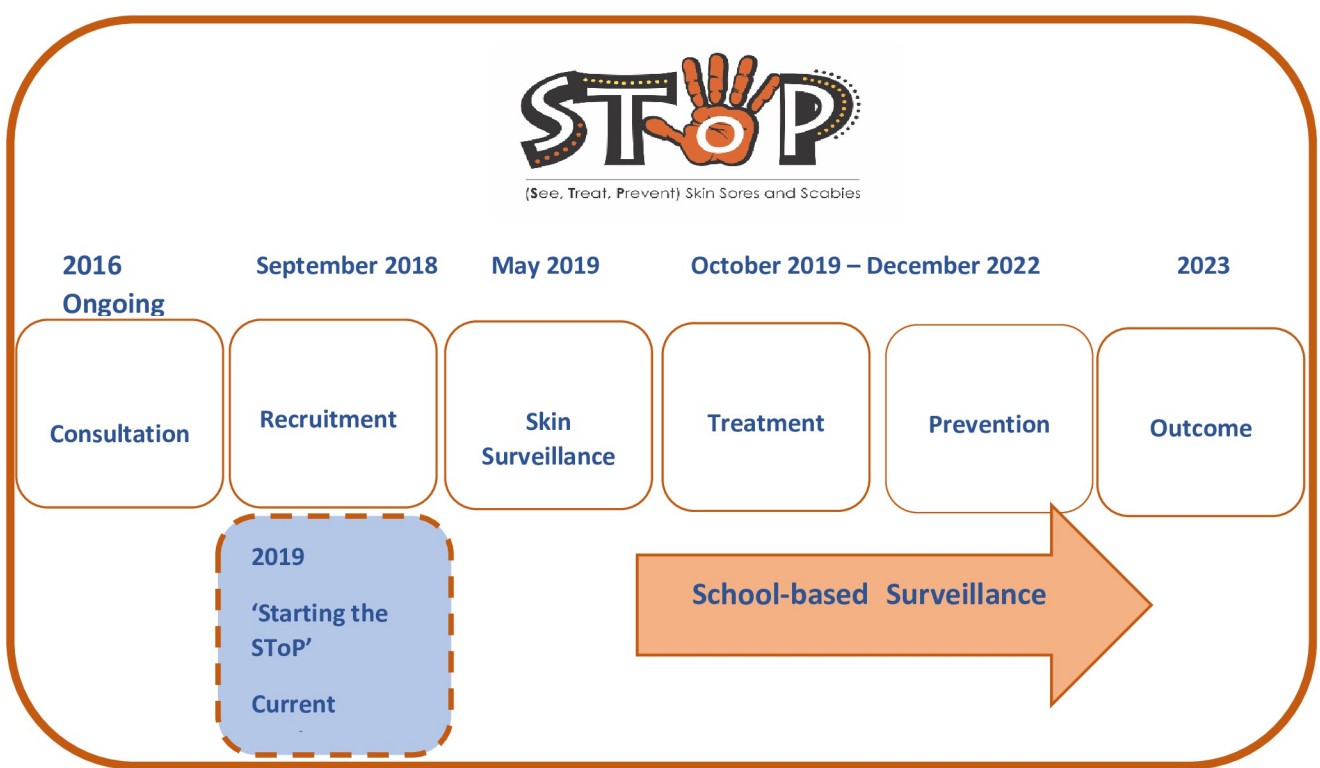

**Fig 1. SToP trial and current research project.**

way (reciprocal) learning approaches and involvement of Aboriginal people in all elements of the research process [10]. Two-way learning shifts power dynamics to strengthen Aboriginal governance and leadership as well as strengthening non-Aboriginal researchers' capacity, and commitment to gain greater knowledge and understanding of, and respect for Aboriginal culture [10]. A CPAR approach enables decolonisation, knowledge sharing and promoting social change [9, 13], providing an appropriate framework for research involving Australian Aboriginal people.

Recruiting participants from culturally diverse backgrounds into research and clinical trials can be challenging for researchers [17–22]. Fear of judgement and stigma of the research topic can create challenges for recruiters [21–23]. However, building trusting relationships with stakeholders and communities [18, 19, 24, 25], employing Aboriginal people [18, 19] and using culturally appropriate visual resources during recruitment can help overcome these difficulties [20, 26–28]. In Australia, it is an ethical requirement to provide consent and information forms in plain English to potential participants during the recruitment process [29–31]. Culturally appropriate pictorial flip charts can also be an effective communication method [26, 28, 32–34].

Despite previous research reporting recruitment barriers and facilitators, there is limited published studies reporting the perspectives of Aboriginal people involved in the recruitment process [31]. Therefore, this process evaluation included staff members' and community member's perspectives to answer the following two questions:

1. What facilitators and barriers impacted recruitment?

2. Was the co-designed flipchart a useful resource for clearly explaining the importance of healthy skin and the rationale of the SToP trial during recruitment?

## Methods

### Study design and theoretical frameworks

This qualitative project acknowledges the importance of Indigenous standpoint theory [15] and seeks to bring this standpoint into the analysis through discussions with Aboriginal researchers and community members. As a non-Aboriginal raised in New Zealand, TM acknowledges the worldview, conscious values, and privilege she brings to this research as lead researcher and is committed to the constructivism philosophy [35]. This has required adoption of a critical reflexive stance and awareness of the enduring effects of colonialism that continues to impact the lives of Aboriginal people today.

A constructivist approach was used in this research as a mechanism to explore the multiple perspectives emerging from the two groups of participants. Enablers and challenges of recruitment into clinical trials has been well documented from an organisational perspective, however Aboriginal participants' perspectives have not been previously considered [31]. The strength of a constructivist approach allows the researcher to understand the factors impacting participants lives that are not always understood from an organisational level when recruiting into research. The interrelationship between an instrumental case study [36], constructivism and symbolic interactionism [37] is integral to changing future recruitment processes that have been a historically dominant worldview. Situating constructivism within an ecological framework helped to reveal the perspectives regarding the enablers and challenges via a different lens.

Instrumental case studies focus on the challenges arising in a case which underpins the principle and purpose of this research [36]. In this instance, the case study encompasses nine remote Aboriginal communities in the Kimberley and the collaborative partnership involved in the SToP recruiting process.

**Selection of participants.**   Purposive sampling [38] was used by TM to recruit via email, staff members who were involved in the SToP recruitment process and able to provide their perspectives on what enablers and barriers impacted SToP recruitment from the organisational level. Community members were approached face-to-face by JJ and asked if they had been involved in the SToP recruitment process. If so, JJ explained the purpose of the current research project and invited them to participant in yarning sessions [39]. While he advised participation was voluntary, he also explained the importance of having their perspectives included in the research to show what works well or doesn't work well when recruiting Aboriginal people into research. Purposive sampling in this instance allowed a heterogenous sample [38] within and between the two different groups of participants who had all been involved in the SToP recruitment process. All participants were provided an information sheet explaining the research and assured anonymity [40] prior to providing their written informed consent.

### Data collection and analysis

Semi-structured interviews [38] with Telethon Kids Institute staff and operational staff from the Aboriginal partner organisation facilitating recruitment were conducted by TM to investigate the enablers and barriers they experienced during the recruitment process. Semi-structured interviews ranged from 30 to 60 minutes in duration and were conducted either face-to-face in Perth, or via telephone for those participants residing in the Kimberley. One-on-one culturally appropriate yarning sessions facilitating cultural safety [41] were conducted face-to-face by JJ with community members in four of the nine SToP communities and ranged from 20 to 90 minutes in duration. Throughout the interviews and yarning sessions, key points and researcher's interpretation of their responses were fed back to participants to ensure these accurately reflected their statements. Semi-structured interview guides were piloted with a

small sample of Telethon Kids Institute staff. Yarning session questions were piloted with two Aboriginal community members prior to being used in this research. Interview and yarning sessions guides attached as Appendix A in S1 File.

All six semi-structured interviews and four of the five yarning sessions were audio recorded using a handheld Olympus audio recorder as a reliable and trustworthy tool to capture the narratives [42]. Audio files were uploaded into NVivo 12, computer-assisted qualitative data analysis software [43] and transcribed verbatim with names removed to de-identify participants and ensure anonymity [40]. Project Officer observation reports (POORs) completed by SToP trial staff during previous fieldtrips to communities were reviewed. Those POORs providing historical context to the recruitment process were also uploaded into NVivo 12 ensuring names were removed to protect confidentiality [40]. Each transcript was assigned a code number to protect participant privacy. An inductive thematic analysis of the data including POORS situated within an ecological framework [44] was conducted to reveal the organisational and community level perspectives. Adhering to the question guide, coding followed the broad topic areas and specific theme codes were added where new themes emerged from the data. An iterative process that involved discussions with the primary supervisor (RW) and member checking with participants, facilitated credibility and dependability of the themes [45].

### Ethics

This qualitative study to examine recruitment processes was approved by the health ethics review committees at the Child and Adolescent Health Service (Approval number RGS0000000584), the Western Australian Aboriginal Health Ethics Committee (Reference number: 819) and the University of Western Australia (Reference RA/4/20/4123).

## Results

### Participants

A total of six female staff and six community members participated in individual interviews and yarning sessions about the consent process for the SToP trial. All staff had undergraduate degrees and four had completed or were currently undergoing post-graduate level education, with one identifying as Aboriginal. Three of the six staff members had previous experience in recruiting participants for research. One male and five female Aboriginal community members participated in yarning sessions. Participants of this study are considered a representative sample.

Table 1: Description of participant demographics and data collection activities for the SToP recruitment evaluation project.

In order to achieve the objectives of the study, the questions were grouped into four topics: facilitators, barriers, the use of the culturally appropriate flipchart, and strategies for effective and culturally responsively engagement with communities in research.

**Table 1. Data collection activities.**

| Group | Description | Interview Participants(n = 6) | Yarning Sessions Participants(n = 6) | Aboriginal (n = 6) | Total Participants |
|---|---|---|---|---|---|
| Organisation Staff | Chief Investigator Program Manager Project Officers Research Assistants Environmental Health Worker | 6 | | 1 | 6 |
| Community Members | Elders, Parents & Carers of children involved in the SToP trial | | 6 | 6 | 6 |
| Total | | | | | 12 |

### Facilitators for SToP trial recruitment

**Collaborative partnerships.** When asked what factors had enabled the SToP recruitment process, all participants in the study believed the partnership between Telethon Kids Institute and the local Aboriginal organisation conducting recruitment was the critical success factor. At the organisational level, staff members believed that the leadership, connectedness, and credibility of the partner organisation was essential to achieving high participation rates during recruitment, whereas community members were more inclined to discuss the respectful way recruitment was facilitated by the local Aboriginal organisation and its staff. This ensured community members were well-informed about the research before consenting their children into the SToP study. Staff members reported this approach had resulted in the recruitment of 740 children (70% of target population) from across the nine remote Aboriginal communities involved in the SToP study. These statistics were derived from SToP trial participant records. Informal feedback from recruiting staff indicated families who declined to participate in the SToP trial felt safe in doing so.

*". . .I think it [is] essential, I just think we would [not] have gotten the consents without [partner organisation]"–Staff Member P1*

*"I highly suspect that the numbers we got have been quite good and we would not have got that amount of participation and that amount of people saying yes had we not had staff members from partner organisation leading this process"–Staff Member P2*

*". . .I would definitely use this process again, I think we can learn from it, and I think we could improve it dramatically. . .I do think that the wisdom of the partners to ask us to use people known to the communities to go to their homes and ask for permission to be in the study, is a principle that I 110% support"–Staff Member P4*

*". . .I think what worked well was the fact that we got a lot [of] consents, in some communities, well over a hundred percent [of predicted numbers], so that worked well"–Staff Member P5*

Community members frequently acknowledged the importance of having an Aboriginal partner organisation leading the recruitment process as they felt the recruiting staff were respectful in their approach which facilitated cultural safety. Community members articulated that the visual flipchart was used during recruitment and believed the community were well informed about the research due to the respectful way in which the community members were approached and the language used by the recruiting staff to describe the research.

*". . .They [partner organisation staff] came knocking on the door [and gave] us information about it and asked us if [like the paper document explained] we wanted to do it, be in the trial or not to be either way. But they [were not] pushy or anything, some people [community members] when they see, especially in communities when they see non-Aboriginal people rock up to the door they just want to shut the door and go back inside, they do [not] want to talk. . ."–Community Member P9*

*"[Yes], we went through that chart [pointing to flipchart], we were all sitting around most of us were mothers and there were other people like [community member name] and [they] were involved in that meeting that recognised things [skin infections]"–Community Member P10*

*"Yeah, they [partner organisation staff] [came] into the school and present[ed] something [for] the kids and some of the parents were there, so they were aware that it was [going to] happen in the community. Some people they just rock up and they want to do all of these*

*things but [provide] no notification or information. [This way of doing it] was really good"–Community Member P9*

*". . .It [SToP trial information] was [not] hard for me to understand [recruiting staff member] used words like Kriol. . ."–Community Member P8*

**Community relationships.** Personal relationships, including direct familial relationships between community members and recruiting staff was identified by four staff and two community members as the critical success factor in the recruitment.

*"I think [recruiting staff member] really drove the whole process. . .and the local knowledge that [recruiting staff member] had and the relationships that [recruiting staff member] had in community was really the only way that we could identify families [not] in the trial, yes, that was really good"–Staff Member P1*

*". . .When the [recruiting] person knew the community really well, then I think we were more successful with those [participation] numbers compared to those where they did not really have that same connection. So, I think, it is two of those things: [having] that connection to the community and also that connection to us that strengthened the recruitment success"–Staff Member P2*

Participants indicated these existing relationships influenced participation rates and accelerated the transfer of information about the SToP trial and consent process on the ground.

*"I think where there was a person leading it [recruitment], who was connected to the community, the consent rates were much higher. I think what [was] reported anecdotally [was that] it was much easier to communicate the trial. . ."–Staff Member P3*

*". . .I think it worked really well [when] the communities where someone knew the person who was asking permission. . .When there was a relationship in place already, the understanding [of the trial] happened really quickly"–Staff Member P4*

*". . . it's good for Aboriginal [people] like [partner organisation staff] to come out and talk because we know some of the [those] workers too, they [are] countrymen you know. . ."–Community Member P9*

*". . .[it works well] at least having a person there [in community] that gets you [researchers] past that door literally to get you in and. . .it's the first thing so they [community members] get comfortable with [researchers] [community members can say] oh yeah we know that mob you know, [you] just [need] that ice breaker to say this mob [is alright, it is] right talk to them"–Community Member P10*

**Barriers for SToP trial recruitment.** Barriers emerging from the data included: staff perceptions that the consent process was protracted; recordkeeping systems to support recruitment were incompletely developed by the research organisation prior to implementation; limited information sharing occurred between partner organisations due to concerns with privacy legislation compliance; and the stigma associated with skin infections and scabies, identified by community members.

**Protracted process.** This was compounded by the many competing demands of daily life in remote communities.

*". . .It certainly wasn't just [a] walk up [to] a house once and expect there to be a whole family there who was happy to sign over to the SToP Trial. . .[recruiting staff] definitely had to do a lot of visits and it was a lot more work. So we've got people who've consented in September [and] October 2018, and the screening [school surveillance] happened eight months later, so [with] that time lapse, people have forgotten what they've consented to and they [community members] didn't necessarily realise that with [recruiting staff], they were actually consenting for Telethon Kids. . .The community didn't draw that connection, so when we walk[ed] up saying we're doing the SToP trial, people didn't join the dots [between the partnership]. . ."– Staff Member P3*

*". . .I also think that because it took twelve months for the consent to happen, there's so much information and understanding that gets lost over that time, so was there a way that we could have done it faster? Should we have thought about having more people do it more quickly?"– Staff Member P4*

As per POOR reports and the study protocol, the 12-month time allocation to achieve optimal consent was based on an estimated one hour per family discussion using the flip chart and asking parent's permission to consent their children into the study. Due to a range of circumstances including the wet season (December–March), community mobility associated with cultural obligations (lore and sorry business), and school holidays, multiple visits were required to achieve wide community coverage which contributed to the perceived delays raised by some staff. Additional logistical planning, staffing, travel, and financial costs were required to address these circumstances. Despite this, high participation rates (~70%) were achieved.

*"Communities are in sorry business (mourning) and no travel to communities for non-essential services is permitted or appropriate until further notice. [Recruiting staff] will not be visiting these communities for consent collection until the new year starts"–POORs 13/12/2018*

*"[It has been] difficult for the team to get any consents collected over the holidays as most people had left community, so the process of door-to-door wasn't working."–POORs 31/01/2019*

*"reported by [recruiting staff] via email, collecting consent has been really slow over the wet season and school holidays as people are away from community. . ."–POORs 25/01/2019*

*". . .[Next time we should] do it [recruitment] really, really targeted [I] probably would've done it a whole lot closer to the actual time of the intervention. . .to the time of the trial starting, 'cause there's been a huge gap so I probably would've thought a better way to approach [recruitment] is to have a team that can spend a week in one community, like a week prior to the actual visit [intervention] starting or something like that and really intensely do it, rather than having it dragged out by I think one or two people for a whole year"–Staff Member P3*

At the organisational level, despite substantial planning, staff articulated that the time from initial recruitment to study commencement was prolonged creating flow-on challenges. The POOR reports show it was difficult for the coordinating staff to anticipate the consequence of having an 8-month time period between the start of recruitment and commencement of the actual trial. This relatively protracted recruitment timeframe impacted on the connectivity with, immediate relevance of and reception for the SToP trial when it commenced in May 2019.

**Complex logistics relating to paperwork.** Staff identified administrative and record keeping challenges initially experienced in conducting and monitoring of the consent process.

Staff described the difficulties of collecting consents and recruiting participants in a remote fieldwork context including misunderstandings in communication and lack of clear record sharing protocols between partners.

> "It's tricky at times because there are not necessarily house numbers or you don't know exactly who signs off on forms and dates of birth are not always known for children, and so some of those items that are usually pretty easy to fill out in research when you're not in these environments on a consent form and that kind of thing were much more difficult. So that affected logistics and made it a bit tricky for those on the ground and for us interpreting the forms that we got back, at times, but I think everyone did a great job at getting as much information as possible. . ."–Staff Member P2

> "So we didn't have good systems in place for how we [were] going to get the consent forms, what we [were] going to do with them when we [received] them, how will we [were going] to identify the houses that [recruiting] had already [visited]"–Staff Member P3

> "I think there was a bit of miscommunication"–Staff Member P5

Staff members indicated that the unavailability of community house maps and privacy concerns meant that it was difficult for research and recruiting staff to access or share accurate information about children resident in the communities to cross-reference for consent completeness.

> ". . .so I think that [knowing what houses had been visited] was probably challenging for all of the [partner organisation] staff as well our own staff, we just were [not] sure had people been asked and declined or had they never been asked or had they been visited and weren't home?"–Staff Member P4

> ". . .[there was a] misunderstanding [around] recordkeeping, [and if] records were supposed to be kept. . .partly because [of the] concern around the privacy. . .many communities don't have a house map per se so that you can document on a map or something visual, [so we were unable to confirm what] houses have already been visited and declined or consented"–Staff Member P4

**Learning and refining the processes in an iterative manner.** When asked what challenges were experienced during recruitment, Telethon Kids Institute staff explained that they did not always understand the partner organisation's priorities and challenges with facilitating recruitment in a remote context. However, with ongoing review and reflection by staff throughout the recruitment phase, the development of systems to improve record keeping processes improved over time. Problem solving and two-way learning together by staff in both organisations enabled improvements to be implemented to address challenges.

> ". . .I think that there was the ability to keep on redirecting and reframing and improving the process [that] happened throughout the twelve months. I think it got better and better and probably faster and faster"–Staff Member P4

**Stigma and fear of judgment.** The SToP trial has a clear focus on seeing, treating, and preventing skin infections. Initially some community members were concerned with the stigma that can be associated with skin infections and scabies for fear of being judged.

*"The word scabies [is an issue] when describing the trial as people feel it is an insult to their personal hygiene"–POORs 31-01-2019*

*". . . there are people who [do not] want to participate as they do [not] have scabies and did [not] want to be labelled as having scabies. [We should, therefore, refocus the trial from a positive perspective and explain] to people that we want to know about everyone's skin health, including the people who do not have scabies/sores. That will help us to figure out what some people do differently, why some people get these problems and other people do [not]"–POORs 26-02-2019*

The POORs and community members indicated that some families were reluctant to provide consent as they may have been seen by others as having a dirty house or being unclean. In addition, there was a fear of judgement experienced amongst some community members relating to how they look after their children. This highlighted the need for a sensitive approach from researchers, as well as reframing the research to a strengths-based healthy skin focus.

*"[There is] the need for a strengths-based approach to the language we use for the trial. When we describe it as a skin sore/scabies trial, people think it is only for people with skin sores and scabies. If we call it a healthy skin trial, then it applies to everyone"–POORs 26-02-2019*

*"I gave out a few consent forms for [recruiting staff member] and people [were] still reluctant to actually participate in the program thinking that [people] might have been judging them for not looking after their kids or whatever. . ."–Community Member P10*

*". . .Locals think that we [organisations] are scrutinising the way they live, or we are judging them or whatever, so they put up all these defences and barricades. . ."–Community Member P10*

**Flip chart used in SToP trial recruitment.** The pictorial flip chart, an intervention designed by the SToP trial research team and modified with feedback from StoP trial stakeholders, partners, and clinic staff was used by the recruiters as a visual resource to explain skin infections and the implementation of the StoP trial. While the data did not confirm if the flip chart had been consistently used during recruitment into the StoP trial, community members affirmed it had been useful for clearly and appropriately explaining skin infections.

*"[No] not that particular chart but I think they had the paper [participant information sheet]"–Community Member P8*

*"These are good [pointing to flipchart] if you [are] doing one on one stuff because visual is always good for us mob, [it is] really plain and simple, people can relate to that, [they can say], oh, we've seen that sore on our kid but we didn't know what it was"–Community Member P10*

Staff described how the flip chart was developed using a co-design process to determine content and materials.

*"So the flip chart was always in place as a concept from the beginning, but over time, it was morphed and modified using feedback from the clinic, the research team, the partners, the Aboriginal stakeholders, and different people who have informed what that flip chart looks like"–Staff member P4*

Staff also articulated how the flip chart had been effective for explaining to the children how and why skin and throat swabs were going to be taken during school surveillance visits.

*". . .So we had a [flipchart] and it's good for that [explaining skin infections and surveillance to the children], we always had it in the room so there were pictures of people swabbing and taking temperatures, so if the kids kind of looked a bit unsure, we would show them the picture and explain. . ."–Staff P1*

## Strategies for future recruitment

Staff and community members were asked to reflect on strategies to improve the research recruitment process. Strategies for future research suggested by participants responsible for operational aspects of recruitment included employment and training of local community members in research topics and processes so they feel confident to conduct recruitment and prioritising the joint identification of teams and partnerships through team t-shirts and posters.

**Local community members.** When asked what strategies might improve future recruitment in remote Aboriginal communities, staff members believed employing local community members to conduct recruitment was a critical aspect for recruitment.

*". . .Having someone and upskilling people locally on the ground or finding someone who actually lives in that community would probably be really helpful. . .–Staff Member P2*

*"I think it would have been a much more effective strategy to use community members"–Staff Member P3*

*"So [I] would do it the same, I hope that when we get to do the next study, we will have a much richer web of knowledge and engagement with the communities [so] there are people in place who we can readily employ to do this kind of work. . .[That will be] good for them [community members] because it [is] employment and capacity-building, and [it is] involved in informing information about their families"–Staff Member P4*

*"I think the best way moving forward would be for community members to have a good broad understanding of research, and then to have a particular training in the project that they are then employed to go and get the consents for. What we need in the future moving forward [is] for community members to feel able to take on this role"–Staff Member P5*

Three out of six community members also reiterated the importance of utilising local people for the process as this would contribute to self-determination and empowerment to facilitate positive and sustained changes to the health and wellbeing of families in their communities.

*". . . I reckon working with the community, having people in the community do that, I think. . . [I would] like to see more and more of our people in the medical industry like having nurses and doctors that are Aboriginal people. We [have to] be proud [have] a bit of self-determination there and be happy about it, see someone else doing something good with their lives"– Community Member P8*

*"If you got someone who lives on community as a navigator going to do [recruitment], it [will] only take a day or really half a day 'cause they know where everybody is and then you [are]*

*saying we could do [it] as an individual way of approaching so everybody gets touched on but we could also do it on a collective community meeting as well"–Community Member P9*

*"In the community [a local person who can tell community members] [the organisation] is all right, it's fine we can do this, it's only about this [research topic] we [are] not talking about how you mind your kids we [are] just telling you, [to] give you this information".–Community Member P10*

**Team branding.** Promoting the SToP trial and partnership between the organisations emerged as a key issue within these communities and strategy for future projects. The separate roles the partners played–with the local community organisation leading recruitment and Telethon Kids Institute staff implementing the SToP trial, was not readily recognised, or understood by community members. Study participants suggested that the recruitment process would have been more effective if branding and identification of the partnership had existed. Both staff and community members suggested that having SToP trial t-shirts and posters to promote the teams and partnerships would have helped community members become familiar with research teams visiting their communities throughout the study.

*". . .I think maybe having a mixed team and more shared branding. I think we got the idea right of not having the actual research team doing it [recruitment] ourselves, I think having that little bit of separation [but] I think maybe we were separated a bit too far though"–Staff Member P3*

*"I think what we did [not] really appreciate was any form of branding that might be needed, so we did [not] badge the people with a T-shirt that said we [are] part of the SToP trial and we [are] aligned with Telethon Kids. So, I think that it probably relied on the skills and talents of the individuals who were employed by [partner organisation] much more so than the support that either of the brands [were] able to give. And, [we could] probably learn from that next time. . . creating a T-shirt that was for the team who was delivering the consent"–Staff Member P4*

*"But when you come out you send a picture of your whole team, your photos, so before you even get here they're [community members] familiar with all these posters, [they will say] oh, this one [person is] coming and they [will] see you. . . that's how we are, that's how Aboriginal people are. They [will] see but if they're familiar seeing your face around, if you put the poster up for a week and when you rock up 'cause that's the problem half the time—who are you, you know? So just a little profile of your little team"–Community Member P14*

## Discussion

### Facilitators

This project has demonstrated the value of adopting empowering and participatory methodological approaches [8, 10, 15] throughout research. The project's methods encompassed constructivism [35], Indigenous research methods [15], and CPAR [13]; situating the interpretation and analysis of findings within an ecological framework allowing multiple worldviews to emerge. Importantly, our research privileges voices from Aboriginal community members regarding how recruitment should be conducted in their communities. This transcends traditional methods of research as participants involved in the study are living in communities where historically health research has been conducted without their engagement and often with little benefit to them [8, 9, 15]. Fundamental to the de-colonising approach of this

research was the two-way learning relationship [10] between the student researcher TM and Aboriginal researcher and co-author JJ who shared their knowledges. This two-way learning relationship facilitated a culturally secure [41] environment for yarning sessions where community members were able to openly share their stories with a non-Aboriginal researcher present. The community members' perspectives and experiences reflected in this research, particularly around stigma and fear of judgement, highlight the ongoing legacy of colonisation.

Interviews and yarning circles with staff and community members involved in the SToP trial recruitment process highlight several facilitators and barriers to obtaining participant consent. The key facilitators included the outsourcing of recruitment to an Aboriginal partner organisation with strong cultural ties to the communities involved, on the advice of the Aboriginal leaders in the SToP trial partnership. This approach ensured a culturally appropriate, community-led recruitment process and is a key strength and recurrent theme consistently identified by the evaluation. This enables short-cuts to the process by relying on existing trust-based relationships to secure participation. Community members participating in yarning circles also identified the value of this approach, particularly having Aboriginal people explaining the research to them in a meaningful, easily understandable way. All community members confirmed that recruiting staff communicated the SToP trial effectively creating substantial awareness of the research being conducted in their communities.

The culturally appropriate co-designed flip chart which incorporated both the study logo and logos of all partner organisations was well-received by community members and helped staff to explain skin infections during school surveillance visits. In addition, identification of teams and partnership through 'branding' was also recommended as a strategy. Whilst the SToP trial and partner organisation logos were present on all study paperwork, the staff believed a more visible SToP trial T-shirt would have provided greater recognition of the partnership and the trial.

A strong theme emerging from the data as a critical success factor was having the Aboriginal partner organisation lead the SToP trial recruitment process. Staff attributed the high recruitment rates to the collaborative partnership where the Aboriginal recruiting staff had strong connections with the communities in SToP. Consistent with the literature, recruiting staff with personal connections to communities enhanced research recruitment rates [21, 24, 25]. This suggests that recruitment may meet less resistance in communities where prior personal relationships between recruiting staff and community members existed [21, 24]. Culturally appropriate pictorial flip charts to translate health literacy in a meaningful way to Aboriginal people have been actioned in several different projects and proven to be an effective communication method [26, 28, 32–34]. While the data from this research did not reveal whether the flip chart was consistently used in recruitment or whether it influenced participation into the trial, positive feedback from participants reiterated the visual resource cut through concepts and language barriers to explain about skin infections and the SToP trial.

## Lessons learnt

The importance of stronger research partnerships in which Aboriginal organisations have shared leadership of the project and process, including recruitment has been revealed in this study. We have learned there needs to be a commitment to better governance, partnership building, greater commitment to co-design and equal ownership of continuous improvement through-out the partnership. Formalised partnership pathways should be in place well before recruitment is scheduled so that the local Aboriginal service partner has 'shared lead' for development of the recruitment process as relevant to the specific location and specific study or project.

Conducting recruitment in a remote context can be a complex and protracted process for external researchers [19]. Whilst this is the rhythm of life in the remote Kimberley region of Australia, much of the protracted process was predictable. As identified by staff, there is a need for a more targeted and coordinated approach to conduct recruitment closer to the implementation of the intervention for continuity and relationship building. Recognising the possibility of substantial delays when conducting research with remote Aboriginal communities, a well-constructed engagement strategy to keep the project relevant and contextual whilst navigating research ethics, governance and recruitment is recommended. Ensuring that funding applications include budget to do this activity appropriately is foundational to success and should be included in the very early stages of co-design discussions.

The limited administrative organisation, unclear documentation, and maintenance of recordkeeping systems during the initial recruitment phase was a strong theme emerging from the data. Unavailability of community street maps and privacy concerns made it difficult for the Aboriginal researchers to accurately monitor which houses had been visited and whether families had consented. Initially, the lack of a methodical approach to paperwork logistics created some confusion and miscommunication between the partners. However, as part of the CPAR approach regular project implementation meetings were conducted where staff discussed recruitment issues and how to address them. Staff reflected on how houses could be more effectively tracked through a more systematic recording of data while still respecting privacy. This resulted in collectively developing a more systematic approach to paperwork logistics which improved the recruitment process over time. These challenges provide insight into the complexities and realities of conducting research in a remote context, identifying the importance of transferring data capture skills from research staff to partner organisations and of collaborative partnerships resolving issues together. While valuable lessons were learnt for the SToP trial, highlighting these challenges, and developing strategies to address them significantly adds to the literature for future research projects conducted in remote Aboriginal communities.

Fear of judgment and stigma pertaining to health topics is not a new concept in research [21–23, 46] but an important one for researchers to consider. In Australia, the enduring effects of colonialism continually impacting Aboriginal individuals, families and communities [11], underly the stigma and fear of judgment surrounding skin infections and scabies in children that emerged as a concern. This reiterates the need for researchers and health professionals to reassure individuals that the study focus is about healthy skin. Acknowledgment of the enduring effects of colonialism requires broaching research topics with sensitivity and non-judgment [21, 22, 46]. The findings from the SToP trial support these recommendations where recruiting staff articulated the information to community members using a positive strength-based approach focusing on healthy skin for everyone.

## Future strategies

Participants in this research emphasised a future strategy of employing and training local community members on research projects to build trust and relationships between researchers and community members. Employing well-respected community members who can share their knowledge and provide cultural guidance will be an essential element of Telethon Kids Institute research projects in the future.

Promoting partnerships through a recognised brand within the community and between the organisations emerged as a key issue and strategy for future projects. To help community members identify teams and partnerships, it was suggested that branded t-shirts and posters be used. Consistency and familiarity are important aspects of building trust and relationships

between researchers and communities as well as maintaining visibility of the SToP trial over a prolonged period.

## Implications for public health

We contextualise staff reports on recruitment with the richness and privileging of Aboriginal community voices in the SToP trial recruitment process. Further, this project validates the cultural appropriateness of using pictorial flip charts to explain research and translate Western health concepts to Aboriginal people [28, 33, 34]. The collaborative recruitment approach used in the SToP trial is novel in a remote clinical trial context focusing on skin health. While the SToP trial recruitment was a protracted process that created administrative challenges and raised concern of stigma around scabies and skin infections, the findings support the effectiveness of having a well-respected local Aboriginal organisation lead recruitment and employing local community members as navigators [24, 25]. The findings reported here provide strategies to help inform best practice in future research engaging Aboriginal people in remote and clinical trial contexts.

## Strengths and limitations

This research project was designed by an early non-Indigenous career researcher with guidance and support from supervisors and Kulunga Aboriginal Research Development Unit team staff in Perth and Broome. Having a local Aboriginal mentor to provide cultural guidance throughout the process was a key strength. While only a small sample participated in the study, the rich narratives resulting from the semi-structured interviews and yarning sessions confirmed by the POORs align with the case study design. This design underpinned by Constructivism and Indigenous research methods embedded within an ecological framework has privileged Aboriginal peoples' worldviews, to ensure they have a genuine voice in this research, and to provide insightful strategies for future research for themselves and other Aboriginal communities.

## Conclusions and recommendations for future research

The culturally responsive and highly effective approach to the SToP recruitment process validate CPAR and constructivism principles of having Aboriginal people involved in all elements of research conducted in their communities, to ensure Aboriginal worldviews underpin the research. Establishing strong collaborations well before recruitment is scheduled ensures equity in the process and all phases of the research are specific to the location of the study and the research is warranted. Embedding these collaborations within a CPAR approach facilitated a productive space for a shared iterative dialogue where reflecting on the research process helps to continually recognise where changes or improvements can be made.

Practical learnings from our research reveal mechanisms such as iPads for capturing data and streamlining processes may improve record keeping and awareness of how recruitment is progressing, rather than hard copy paper documents used for SToP recruitment. Establishing strong and genuine partnerships outlining clear protocols including confidentiality, data sharing and role expectations and responsibilities through formal partner agreements or Memorandums of Understanding can help mitigate logistical challenges of conducting research in a remote context. Budgeting for the cost and time of recruitment in partnership and using branding such as t-shirts can help maintain awareness prior to the commencement of the study. Further, our commitment to acknowledging and disseminating the findings and recommendations from this research and maintaining collaborative relationships between partners will inform the implementation phase of the SToP trial.

## Supporting information

**S1 File. Question guides.**
(DOCX)

**S2 File. Table outlining how themes emerged.**
(DOCX)

**S3 File. Minimum data set.**
(DOCX)

## Acknowledgments

The authors acknowledge all SToP trial communities, stakeholders, partners and RACP. The authors thank all the participants for sharing their experiences.

## Author Contributions

**Conceptualization:** Tracy McRae, Roz Walker.

**Data curation:** Tracy McRae, John Jacky.

**Formal analysis:** Tracy McRae, John Jacky.

**Investigation:** Tracy McRae, John Jacky.

**Methodology:** Tracy McRae, Roz Walker, Asha C. Bowen.

**Project administration:** Tracy McRae.

**Supervision:** Roz Walker, Judith M. Katzenellenbogen, Juli Coffin, Jonathan Carapetis, Asha C. Bowen.

**Validation:** Jonathan Carapetis.

**Writing – original draft:** Tracy McRae.

**Writing – review & editing:** John Jacky, Judith M. Katzenellenbogen, Juli Coffin, Ray Christophers, Asha C. Bowen.

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
