## [Decision Letter · Decision Letter 0]

20 Jun 2022

PONE-D-21-15498

Starting the SToP: Lessons Learnt from a Collaborative Research Approach

PLOS ONE

Dear Dr. McRae,

Thank you for submitting your manuscript to PLOS ONE. After careful consideration, we feel that it has merit but does not fully meet PLOS ONE’s publication criteria as it currently stands. Therefore, we invite you to submit a revised version of the manuscript that addresses the points raised during the review process.

 The progress was plodding as the editorial board has struggled to get the manuscript reviewed since its submission. As an editor, I would expect you to understand the circumstances.

We look forward to receiving your revised manuscript.

Kind regards,

M Tanveer Hossain Parash

Academic Editor

PLOS ONE

Journal Requirements:

2. Thank you for stating in the text of your manuscript "All participants provided written informed consent prior to participating in this research." Please also add this information to your ethics statement in the online submission form.

3. Please include a copy of the interview guide used in the study, in both the original language and English, as Supporting Information, or include a citation if it has been published previously.

4. In your Methods section, please provide additional information about the participant recruitment method and the demographic details of your participants. Please ensure you have provided sufficient details to replicate the analyses such as: 

a) the recruitment date range (month and year), 

b) a description of any inclusion/exclusion criteria that were applied to participant recruitment, 

c) a table of relevant demographic details, 

d) a statement as to whether your sample can be considered representative of a larger population, and 

e) a description of how participants were recruited.

6. Thank you for stating the following in the Acknowledgments Section of your manuscript: "The authors acknowledge all SToP trial communities, stakeholders, partners and RACP. The authors thank all the participants for sharing their experiences. Funding was received from the National Health and Medical Research Council [NHMRC] (project grant 1128950), Health Outcomes in the Tropical NORTH [HOT NORTH 113932] Indigenous Capacity Building Grant), and WA Health Department and Healthway grants contributed to this research. ACB receives a NHMRC investigator Award (1175509)."

Please remove any funding-related text from the manuscript and let us know how you would like to update your Funding Statement. Currently, your Funding Statement reads as follows: "ACB - National Health and Medical Research Council [NHMRC] (project grant 1128950)

https://www.nhmrc.gov.au/

ACB - Health Outcomes in the Tropical NORTH [HOT NORTH 113932] Indigenous Capacity Building Grant)

https://www.hotnorth.org.au/

ACB - Western Australia Government Healthway Grant 33088

https://www.healthway.wa.gov.au/our-funding/

ACB NHMRC investigator Award (1175509)

https://www.nhmrc.gov.au/

7. In your Data Availability statement, you have not specified where the minimal data set underlying the results described in your manuscript can be found. PLOS defines a study's minimal data set as the underlying data used to reach the conclusions drawn in the manuscript and any additional data required to replicate the reported study findings in their entirety. All PLOS journals require that the minimal data set be made fully available. For more information about our data policy, please see http://journals.plos.org/plosone/s/data-availability.

Additional Editor Comments (if provided):

Following are the required corrections from the editor:

1. Please remove the sentence "This study was undertaken by a student researcher (TM) as part of a Master’s dissertation at the University of WA."

2. Please re-write the sentence " As a non-Aboriginal raised in New Zealand, TM acknowledges the worldview, conscious values, and privilege she brings to this research and is committed to the constructivism philosophy39" replacing the initial of the researcher (TM) by the phrase "the researcher".

3. Please remove the following sentences from the methodology section "Throughout this research, TM received cultural guidance, knowledge and mentoring from Aboriginal co-authors, John Jacky (JJ), Juli Coffin (JC), and Ray Christophers (RC). The three non- Aboriginal co-authors provided mentorship and supervision as follows; qualitative research knowledge Roz Walker (RW); project planning Judy Katzenellenbogen (JK), understanding of skin infections Asha Bowen (AB). It is with their ongoing guidance and support; this research was possible."

4. Please re-write the acknowledgement following the author’s guidelines for financial disclosure.

Please refer to the attachment file for detailed comments (Reviewer 2).

Reviewers' comments:

Reviewer's Responses to Questions

**Comments to the Author**

1. Is the manuscript technically sound, and do the data support the conclusions?

Reviewer #1: Yes

Reviewer #2: Yes

2. Has the statistical analysis been performed appropriately and rigorously? 

Reviewer #1: N/A

Reviewer #2: N/A

3. Have the authors made all data underlying the findings in their manuscript fully available?

Reviewer #1: Yes

Reviewer #2: No

4. Is the manuscript presented in an intelligible fashion and written in standard English?

Reviewer #1: Yes

Reviewer #2: Yes

5. Review Comments to the Author

Reviewer #1: The authors have submitted a well-structured paper with sound methodology. The results lead to constructive suggestions for conducting future research in remote communities, especially Indigenous communities. The study highlights a ‘disconnect’ between institutional research imperatives and community perspectives. It underlines the importance of community partnerships, especially when it comes to research study recruitment. The abstract is clear and adequately outlines the research process. This is a qualitative study and the sample size is pretty small. Nevertheless, the interviews have produced some rich data. The authors analysed the data in an appropriate manner and presented fitting results. They then provided a comprehensive discussion and came up with some sound recommendations.

It would have been helpful to have page numbers and the authors should include them for final submission. The paper needs shortening by 40%, particularly the introduction and the discussion. This shouldn’t be too difficult, without loss of meaning or message, aiming to trim away any repetition. For example the authors could prune the first two paragraphs of the methods section to half the size, taking care to clarify any rarefied concepts and jargon. Also, the authors would improve the paper’s readability by editing the text, making sure that the verb appears towards the front of each sentence. Putting the verb at the end usually makes for turgid reading. Lastly, in the words of the great Gordon Guyatt: “Conduct one edit of your paper in which your only goal is to change passive to active voice.” [Guyatt GH, Haynes RB. J Clin Epidemiol 2006;59:900-6]. That would certainly help.

Reviewer #2: Dear Authors

Thank you for giving me the opportunity to read and review your work. It is an interesting topic. As qualitative researcher I can imagine how you went through the process. My intention is to improve the quality of the research, thus my comments:

Overall, there is novelty. It is a well written manuscript. However, improvement is necessary. My main concerned is that the manuscript lack technical information, hence this need to be included for example the process of analysis. How you derived the themes. This is crucial for reader to recognize the scientific process of doing the research. I have returned your manuscript with my comments. I hope that helps to improve the manuscript. Good luck and wish you all the best.

6. PLOS authors have the option to publish the peer review history of their article (what does this mean?). If published, this will include your full peer review and any attached files.

Reviewer #1: No

Reviewer #2: **Yes: **Assoc. Prof. Dr. Zaleha Othman

---

## [Author Response · Author response to Decision Letter 0]

3 Aug 2022

As lead author and on behalf of all co-authors, I would like to thank the reviewers for their feedback and comments and allowing us to reconsider elements of the manuscript and make required amendments. We believe we have addressed all comments accordingly and provided additional information and context where necessary. Please see author responses bolded below. 

1. Please ensure that your manuscript meets PLOS ONE's style requirements, including those

for file naming. The PLOS ONE style templates can be found at

https://journals.plos.org/plosone/s/file?

id=wjVg/PLOSOne_formatting_sample_main_body.pdf and

https://journals.plos.org/plosone/s/file?

id=ba62/PLOSOne_formatting_sample_title_authors_affiliations.pdf

1. Response: Manuscript checked to meet PLOS ONE’s style requirements.

2. Thank you for stating in the text of your manuscript "All participants provided written

informed consent prior to participating in this research." Please also add this information to

your ethics statement in the online submission form.

2. Response: Statement added to ethics statement in the online submission form

3. Please include a copy of the interview guide used in the study, in both the original language

and English, as Supporting Information, or include a citation if it has been published

previously.

3. Response: Interview and yarning session guide in English (no original language) included as Appendix A.

4. In your Methods section, please provide additional information about the participant

recruitment method and the demographic details of your participants. Please ensure you have

provided sufficient details to replicate the analyses such as:

b) a description of any inclusion/exclusion criteria that were applied to participant

recruitment,

c) a table of relevant demographic details,

d) a statement as to whether your sample can be considered representative of a larger

population, and

e) a description of how participants were recruited.

4. Response: 

a. Description of participants included

b. Table of demographic details included

c. Statement as to whether sample can be considered representative included

d. Description of how participants were recruited included 

5. We note that the grant information you provided in the ‘Funding Information’ and

‘Financial Disclosure’ sections do not match.

When you resubmit, please ensure that you provide the correct grant numbers for the awards

you received for your study in the ‘Funding Information’ section.

5. Response: Funding Information section updated 

"The authors acknowledge all SToP trial communities, stakeholders, partners and RACP. The

authors thank all the participants for sharing their experiences. Funding was received from the

National Health and Medical Research Council [NHMRC] (project grant 1128950), Health

Outcomes in the Tropical NORTH [HOT NORTH 113932] Indigenous Capacity Building

Grant), and WA Health Department and Healthway grants contributed to this research. ACB

receives a NHMRC investigator Award (1175509)."

We note that you have provided funding information that is not currently declared in your

Funding Statement. However, funding information should not appear in the Acknowledgments

section or other areas of your manuscript. We will only publish funding information present in

the Funding Statement section of the online submission form.

Please remove any funding-related text from the manuscript and let us know how you would

like to update your Funding Statement. Currently, your Funding Statement reads as follows:

"ACB - National Health and Medical Research Council [NHMRC] (project grant 1128950)

https://www.nhmrc.gov.au/

ACB - Health Outcomes in the Tropical NORTH [HOT NORTH 113932] Indigenous

Capacity Building Grant)

https://ddec1-0-en-ctp.trendmicro.com:443/wis/clicktime/v1/query?

url=https://www.hotnorth.org.au&umid=2c0a3bb6-1437-4835-826b-

975f0e4c70e6&auth=bd49bbd20ffeb7d6acc8e9a85bb2e9a8f7a67034-

f09cbc8d55e158912bf138eb9fafca0e90569062

ACB - Western Australia Government Healthway Grant 33088

https://www.healthway.wa.gov.au/our-funding/

ACB NHMRC investigator Award (1175509)

https://www.nhmrc.gov.au/

The funders had no role in study design, data collection and analysis, decision to publish, or

preparation of the manuscript."

Please include your amended statements within your cover letter; we will change the online

submission form on your behalf.

6. Response: Funding information has been removed from acknowledgements section and added to cover letter. 

7. In your Data Availability statement, you have not specified where the minimal data set

underlying the results described in your manuscript can be found. PLOS defines a study's

minimal data set as the underlying data used to reach the conclusions drawn in the manuscript

and any additional data required to replicate the reported study findings in their entirety. All

PLOS journals require that the minimal data set be made fully available. For more information

about our data policy, please see http://journals.plos.org/plosone/s/data-availability.

Upon re-submitting your revised manuscript, please upload your study’s minimal underlying

data set as either Supporting Information files or to a stable, public repository and include the

relevant URLs, DOIs, or accession numbers within your revised cover letter. For a list of

acceptable repositories, please see http://journals.plos.org/plosone/s/data-availability#locrecommended-

repositories. Any potentially identifying patient information must be fully

anonymized.

Important: If there are ethical or legal restrictions to sharing your data publicly, please explain

these restrictions in detail. Please see our guidelines for more information on what we consider

unacceptable restrictions to publicly sharing data: http://journals.plos.org/plosone/s/dataavailability#

loc-unacceptable-data-access-restrictions. Note that it is not acceptable for the

authors to be the sole named individuals responsible for ensuring data access.

We will update your Data Availability statement to reflect the information you provide in your

cover letter.

7. Response: Minimal data set uploaded as Supporting Information File. 

8. Please review your reference list to ensure that it is complete and correct. If you have cited

papers that have been retracted, please include the rationale for doing so in the manuscript

text, or remove these references and replace them with relevant current references. Any

changes to the reference list should be mentioned in the rebuttal letter that accompanies your

revised manuscript. If you need to cite a retracted article, indicate the article’s retracted status

in the References list and also include a citation and full reference for the retraction notice.

Additional Editor Comments (if provided):

8. Response: Reference list reviewed & correct. 

Following are the required corrections from the editor:

1. Please remove the sentence "This study was undertaken by a student researcher (TM) as

part of a Masters’ dissertation at the University of WA."

Response: Sentence Removed 

2. Please re-write the sentence " As a non-Aboriginal raised in New Zealand, TM

acknowledges the worldview, conscious values, and privilege she brings to this research and is

committed to the constructivism philosophy39" replacing the initial of the researcher (TM) by

the phrase "the researcher".

Response: Sentence Reworded 

3. Please remove the following sentences from the methodology section "Throughout this

research, TM received cultural guidance, knowledge and mentoring from Aboriginal coauthors,

John Jacky (JJ), Juli Coffin (JC), and Ray Christophers (RC). The three non-

Aboriginal co-authors provided mentorship and supervision as follows; qualitative research

knowledge Roz Walker (RW); project planning Judy Katzenellenbogen (JK), understanding of

skin infections Asha Bowen (AB). It is with their ongoing guidance and support; this research

was possible."

Response: Removed from methodology section. 

4. Please re-write the acknowledgement following the author’s guidelines for financial

disclosure.

Response: Supervisors not added to financial disclosure. 

Please refer to the attachment file for detailed comments (Reviewer 2).

Reviewers' comments:

Reviewer's Responses to Questions

Comments to the Author

1. Is the manuscript technically sound, and do the data support the conclusions?

The manuscript must describe a technically sound piece of scientific research with data that

supports the conclusions. Experiments must have been conducted rigorously, with appropriate

controls, replication, and sample sizes. The conclusions must be drawn appropriately based on

the data presented.

Reviewer #1: Yes

Reviewer #2: Yes

2. Has the statistical analysis been performed appropriately and rigorously?

Reviewer #1: N/A

Reviewer #2: N/A

3. Have the authors made all data underlying the findings in their manuscript fully available?

The PLOS Data policy requires authors to make all data underlying the findings described in

their manuscript fully available without restriction, with rare exception (please refer to the

Data Availability Statement in the manuscript PDF file). The data should be provided as part

of the manuscript or its supporting information, or deposited to a public repository. For

example, in addition to summary statistics, the data points behind means, medians and

variance measures should be available. If there are restrictions on publicly sharing data—e.g.

participant privacy or use of data from a third party—those must be specified.

Reviewer #1: Yes

Reviewer #2: No

4. Is the manuscript presented in an intelligible fashion and written in standard English?

PLOS ONE does not copyedit accepted manuscripts, so the language in submitted articles

must be clear, correct, and unambiguous. Any typographical or grammatical errors should be

corrected at revision, so please note any specific errors here.

Reviewer #1: Yes

Reviewer #2: Yes

5. Review Comments to the Author

Please use the space provided to explain your answers to the questions above. You may also

include additional comments for the author, including concerns about dual publication,

research ethics, or publication ethics. (Please upload your review as an attachment if it exceeds

20,000 characters)

Reviewer #1: The authors have submitted a well-structured paper with sound methodology.

The results lead to constructive suggestions for conducting future research in remote

communities, especially Indigenous communities. The study highlights a ‘disconnect’ between

institutional research imperatives and community perspectives. It underlines the importance of

community partnerships, especially when it comes to research study recruitment. The abstract

is clear and adequately outlines the research process. This is a qualitative study and the sample

size is pretty small. Nevertheless, the interviews have produced some rich data. The authors

analysed the data in an appropriate manner and presented fitting results. They then provided a

comprehensive discussion and came up with some sound recommendations.

It would have been helpful to have page numbers and the authors should include them for final

submission. The paper needs shortening by 40%, particularly the introduction and the

discussion. This shouldn’t be too difficult, without loss of meaning or message, aiming to trim

away any repetition. For example the authors could prune the first two paragraphs of the

methods section to half the size, taking care to clarify any rarefied concepts and jargon. Also,

the authors would improve the paper’s readability by editing the text, making sure that the

verb appears towards the front of each sentence. Putting the verb at the end usually makes for

turgid reading. Lastly, in the words of the great Gordon Guyatt: “Conduct one edit of your

paper in which your only goal is to change passive to active voice.” [Guyatt GH, Haynes RB. J

Clin Epidemiol 2006;59:900-6]. That would certainly help.

Reviewer #2: Dear Authors

Thank you for giving me the opportunity to read and review your work. It is an interesting

topic. As qualitative researcher I can imagine how you went through the process. My intention

is to improve the quality of the research, thus my comments:

Overall, there is novelty. It is a well written manuscript. However, improvement is necessary.

My main concerned is that the manuscript lack technical information, hence this need to be

included for example the process of analysis. How you derived the themes. This is crucial for

reader to recognize the scientific process of doing the research. I have returned your

manuscript with my comments. I hope that helps to improve the manuscript. Good luck and

wish you all the best.

6. PLOS authors have the option to publish the peer review history of their article (what does

this mean?). If published, this will include your full peer review and any attached files.

If you choose “no”, your identity will remain anonymous but your review may still be made

public.

Do you want your identity to be public for this peer review? For information about this

choice, including consent withdrawal, please see our Privacy Policy.

Reviewer #1: No

Reviewer #2: Yes: Assoc. Prof. Dr. Zaleha Othman

While revising your submission, please upload your figure files to the Preflight Analysis and

Conversion Engine (PACE) digital diagnostic tool, https://pacev2.apexcovantage.com/. PACE

helps ensure that figures meet PLOS requirements. To use PACE, you must first register as a

user. Registration is free. Then, login and navigate to the UPLOAD tab, where you will find

detailed instructions on how to use the tool. If you encounter any issues or have any questions

when using PACE, please email PLOS at figures@plos.org. Please note that Supporting

Information files do not need this step.

In compliance with data protection regulations, you may request that we remove your personal registration

details at any time. (Remove my information/details). Please contact the publication office if you have any

questions.

Response to Reviewer 1 Comments 

Thank you for reviewing our paper and providing your comments. This has allowed me and the co-authors the opportunity to re-address and strengthen the manuscript and reduce the sections below without losing context. 

• Page and line numbers have been added to the manuscript for ease of reading. 

• Introduction, methods, and discussion sections of the manuscript have been restructured with consideration to context.

• Specific lines have been deleted for reduction as outlined below: 

66-69

379-381

462-464

665-668

712-714

720-725

768-774

783-789

797-803

874-879

886-888

899-901

906-908

919-922

951-955

972-982

• Specific lines have been restructured as outlined below. Important to note is the overlap of restructuring to address Reviewer 2 comments also. 

77-88

90-104

106-113

135-152

154-165

183-225



Response to Reviewer 2 Comments 

Thank you for reviewing our paper and providing your comments. This has allowed me and the co-authors to re-address and strengthen the manuscript, provide additional technical information, context on participant recruitment, explaining the data analysis process and how themes emerged. I have replied to your comments in the manuscript in addition to providing more information below.

Reviewer Comment (Line 100)

Who are they? It is suggested that the author described both

Author Response to Reviewer 2

SToP trial partners now named

Reviewer Comment Line 214 June 15

The authors mentioned constructivism, however, the author also mentioned using intrinsic case study.. This is confusing. As to my knowledge one would not suggest using constructivism to explain case study. Case study is more of symbolic interactionist. I hope the authors could justify the use of constructivism to case study.

Author Response (also in Manuscript) 

Thank you for the opportunity to reclarify my thinking around what it is that we actually engaged in with this research. On deeper reading of these methodologies, and in consultation with my supervisors I have redefined the intrinsic case study to an instrumental case study to reflect more accurately that we aimed to collect multiple perspectives regarding the enablers and barriers of the case, rather than focus on the case itself as an Intrinsic case study. It was not the intention of constructivism to be used to justify the case study, a constructivist approach was used to re-examine the interrelationship between constructivism /instrumental case study and symbolic interactionist. We have addressed this between lines 203-212. We hope that you are satisfied we have clarified this confusion. 

Reviewer Comments Line 207 June 15. 

This is good explanation to support the use of intrinsic case study however, the main reason should be the highlight of employing this method. I would suggest that the authors describe the fundamental of intrinsic case study i.e. to provide in-depth understanding of the process ….

The rational of using multiple sources is secondary explanation. The primary use of case study is to provide understanding of certain process and this is not visible in the narrative

Author Response (also in manuscript)

As discussed between lines 203-212 and aligning with the constructivist approach, we have defined an intrinsic case study to a reflect an instrumental case study as we believe this more accurately reflects the aims of the research which was to identify the challenges and enablers of the recruitment process experienced by both group of participants.

Reviewer Comments Line 230 June 15. 

This is supportive of the purposive sampling however, the reason for purposive sampling is not being justified. To improve this, it is suggested that the author explain the criteria of selection. Why were the two groups were chosen? How many people are in each group? How the selection was made (including the criteria). Also, the authors should support the justifications with qualitative gurus such as Silverman, Patton (2002) is a good book to refer to support your justifications

Author Response (also in manuscript)

Purposive sampling was used to recruit participants who had been involved in the recruitment process so they could describe the challenges and enablers. It was important to have two groups, one at the organisational level and the community members who had been involved in the recruiting process. All participants had been involved in some stage of the recruitment process and were able to share their knowledge.

Reviewer Comments Line 245 

What are the duration of the interviews? What are the aim of the interviews? How was it conducted? At the setting? Detail information about the process of interviews should be described.

Author Response (also in manuscript)

The process of the interviews has been described between lines 244-258 including the duration of interviews, aims and how the interviews were conducted. 

Reviewer Comment Line 260

What do you mean ‘with any identifying information removed?

Author Response (also in manuscript)

Addressed in between lines 260-274 how confidentiality was considered by removing names from transcripts 

Reviewer comment Line 277

What are the processes? How was it done to support each of the element of trustworthiness?

Author Response (also in manuscript)

Thank you for the suggestion to read Othman & Hamid paper read which has provided good context for supporting the elements of trustworthiness. Addressed between lines 260-274 and included Othman as reference.

Reviewer comments (Line 311) 

Please provide evidence of how the theme emerged. Reader would want to know the process of the analysis of how the authors derived at these themes

Author Response (also in manuscript)

Addressed between lines 306-322 above, themes emerged from the four topics and coded as barriers, enablers, flipchart, and strategies. From these topics, sub-themes emerged to revealing similarities from participants within the two groups. A table attached as Appendix B showing how themes emerged.

Reviewer comments (Line 311)

Providing quotations as evidence is good. However, it would be best if the authors could provide a table that depicts the analysis process. This is important to show reader how the authors derived at each theme. How was the level of abstraction conducted?

Author Response (also in manuscript)

Addressed between lines 307-323 above, themes emerged from the four topics and coded as barriers, enablers, flipchart and strategies. From these topics, sub-themes emerged to revealing similarities from participants within the two groups. A table attached as Appendix B showing how themes emerged. 

Reviewer comments (Line 317)

What are the nine-communities? It is best if the authors could describe them.

Author Response (also in manuscript)

Nine SToP communities added in line 316.

Reviewer comments (Line 319)

It is suggested that the author provide reference for this as it is a fact.

Author Response (also in manuscript)

Thank you for the suggestion to reference. Deliberating with the SToP trial chief investigator who advised these statistics are part of the participation rates/results internal docs therefore we have not formally referenced, rather just noted for context. 

Reviewer comments (Line 321)

The author claimed that the CPAR emerged as a critical success factor for recruitment. This is interesting. However, it is suggested that the author provide evidence of how it emerged that support this finding?

Author Response (also in manuscript)

Addressed between lines 307-322 is the process of how the CPAR emerged as a critical factor for recruitment. Both groups of participants believed the partnership between TKI and the local organisation conducting recruitment had been successful. From an organisational level, the perspectives focused on the result of high rates of participation whereas from the community level, the focus was on the ‘process’ of the communication and culturally appropriate style of the recruitment that had led to high rates of participation. 

Reviewer comments (Line 364)

How do the community confirmed this fact? It is suggested that the authors provide diagram or figure to show the reader the process of how the community made this confirmation.

Author Response (also in manuscript)

Amended lines 350-355 to provide context aligning with community members quotes & deleting lines 364-366

Reviewer comments (Line 532)

Please justify this statement. What made you said this?

Author Response (also in manuscript)

Reflecting upon the context of this theme, I have restructured lines 525-531 to focus on the iterative process of reframing the process between the collaborative partnership. Lines 532-541 have been deleted entirely.

Reviewer comments (Line 581)

I believe this is the intervention … Please address this as intervention in your narrative

Author Response (also in manuscript)

I have amended lines 582-587 to discuss the flipchart as the intervention used for explaining the SToP trial and skin infections.

Reviewer comments (Line 620)

As I have mentioned above, it is important for the authors to provide the process of how the themes emerged. On that note, it is suggested that the authors provide appendix.

Author Response (also in manuscript)

Thank you for the suggestion to provide an appendix for how themes emerged. Attached as Appendix B is a table outlining how themes emerged from the data. 

Reviewer comments (Line 703)

This is an excellent contribution of this study however I am not able to grasp that in the analysis section/ findings. I would suggest that the authors provide analysis that deliberate this statement.

Author Response (also in manuscript)

The constructivist approach has been addressed in more detail in methodology section and added context in lines 707-712

Reviewer comments (Line 728)

The lesson learned not clearly visible. Perhaps the authors should provide sub section on reflection/ lesson learned

Author Response (also in manuscript)

Moving lines 728-735 and added sub-section on lessons learnt in lines 820 as suggested.

Reviewer comments (Line 938)

Please provide a section on what are the contributions gathered from the findings, practical contribution, theoretical contributions.

Author Response (also in manuscript)

Thank you for your suggestion to provide this section on practical and theoretical contributions. I have restructured lines 939-948 to highlight these contributions. 

Reviewer comments (Line 951)

Perhaps you should be more specific i.e suggest to change to Collaborative participatory

Author Response (also in manuscript)

Restructured this section entirely, deleting lines 951-955 for reduction and repetitiveness

---

## [Decision Letter · Decision Letter 1]

15 Aug 2022

Starting the SToP Trial: Lessons Learnt from a Collaborative Research Approach

PONE-D-21-15498R1

Dear Dr. McRae,

We’re pleased to inform you that your manuscript has been judged scientifically suitable for publication and will be formally accepted for publication once it meets all outstanding technical requirements.

Kind regards,

M Tanveer Hossain Parash

Academic Editor

PLOS ONE

Additional Editor Comments (optional):

Reviewers' comments:

Reviewer's Responses to Questions

**Comments to the Author**

1. If the authors have adequately addressed your comments raised in a previous round of review and you feel that this manuscript is now acceptable for publication, you may indicate that here to bypass the “Comments to the Author” section, enter your conflict of interest statement in the “Confidential to Editor” section, and submit your "Accept" recommendation.

Reviewer #2: All comments have been addressed

2. Is the manuscript technically sound, and do the data support the conclusions?

Reviewer #2: Yes

3. Has the statistical analysis been performed appropriately and rigorously? 

Reviewer #2: Yes

4. Have the authors made all data underlying the findings in their manuscript fully available?

Reviewer #2: Yes

5. Is the manuscript presented in an intelligible fashion and written in standard English?

Reviewer #2: Yes

6. Review Comments to the Author

Reviewer #2: I have reviewed the manuscript and it is satisfactory. The justifications made are acceptable. The manuscript represents an interesting topic and on that note, I would recommended this manuscript to be published.

7. PLOS authors have the option to publish the peer review history of their article (what does this mean?). If published, this will include your full peer review and any attached files.

Reviewer #2: No

---

## [Editor Report · Acceptance letter]

9 Sep 2022

PONE-D-21-15498R1 

Starting the SToP Trial: Lessons from a Collaborative Recruitment Approach 

Dear Dr. McRae:

I'm pleased to inform you that your manuscript has been deemed suitable for publication in PLOS ONE. Congratulations! Your manuscript is now with our production department. 

Kind regards, 

on behalf of

Dr. M Tanveer Hossain Parash 

Academic Editor

PLOS ONE